# PERFORMANCE DISPARITIES BETWEEN ACCENTS IN AUTOMATIC SPEECH RECOGNITION

## ABSTRACT

Automatic speech recognition (ASR) services are ubiquitous. Past research has identified discriminatory ASR performance as a function of racial group and nationality. In this paper, we expand the discussion by performing an audit of some of the most popular English language ASR services using a large and global data set of speech from *The Speech Accent Archive*. We show that, even when controlling for multiple linguistic covariates, ASR service performance has a statistically significant relationship to the political alignment of the speaker's birth country with respect to the United States' geopolitical power. We discuss this bias in the context of the historical use of language to maintain global and political power.

## 1 INTRODUCTION

Automatic speech recognition (ASR) services are a key component of the vision for the future of human-computer interaction. However, many users are familiar with the frustrating experience of repeatedly not being understood by their voice assistant (Harwell, 2018), so much so that frustration with ASR has become a culturally-shared source of comedy (Connell & Florence, 2015; Mitchell, 2018).

Bias auditing of ASR services has quantified these experiences. English language ASR has higher error rates: for Black Americans compared to white Americans (Koenecke et al., 2020; Tatman & Kasten, 2017); for Scottish speakers compared to speakers from California and New Zealand (Tatman, 2017); and for speakers who self-identify as having Indian accents compared to speakers who self-identify as having American accents (Meyer et al., 2020). It should go without saying, but everyone has an accent – there is no "unaccented" version of English (Lippi-Green, 2012). Due to colonization and globalization, different Englishes are spoken around the world. While some English accents may be favored by those with class, race, and national origin privilege, there is no technical barrier to building an ASR system which works well on any particular accent. So we are left with the question, why does ASR performance vary as it does as a function of the global English accent spoken? This paper attempts to address this question quantitatively using a large public data set, *The Speech Accent Archive* (Weinberger, 2015), which is larger in number of speakers (2,713), number of first languages (212), and number of birth countries (171) than other data sets previously used to audit ASR services, and thus allows us to answer richer questions about ASR biases. Further, by observing historical patterns in how language has shifted power, our paper provides a means for readers to understand how ASR may be operating today.

Historically, accent and language have been used as a tool of colonialism and a justification of oppression. Colonial power, originally British and then of its former colonies, used English as a tool to "civilize" their colonized subjects (Kachru, 1986), and their accents to justify their lower status. English as a *lingua franca* today provides power to those for whom English is a first language. People around the world are compelled to promote English language learning in education systems in order to avail themselves of the privilege it can provide in the globalized economy (Price, 2014). This spread of English language may be "reproducing older forms of imperial political, economic, and cultural dominance", but it also exacerbates inequality along neoliberal political economic lines (Price, 2014). In short, the dominance of the English language around the world shifts power in ways that exacerbate inequality.

Further, English is and has historically been used as a nationalist tool in the United States to justify white conservative fears that immigrants pose an economic and political threat to them and has been

used to enforce the cultural assimilation of immigrants (Lippi-Green, 2012). We note that, even within the United States, "Standard American English" is a theoretical concept divorced from the reality of wide variations in spoken English across geographical areas, race and ethnicity, age, class, and gender (Lippi-Green, 2012). As stated in a resolution of the Conference on College Composition and Communication in 1972, "The claim that any one dialect is unacceptable amounts to an attempt of one social group to exert its dominance over another" (Lippi-Green, 2012). The social construct of language has real and significant consequences Nee et al. (2022), for example, allowing people with accents to be passed over for hiring in the United States, despite the Civil Rights Act prohibiting discrimination based on national origin (Matsuda, 1991). Accent-based discrimination can take many forms — people with accents deemed foreign are rated as less intelligent, loyal, and influential (Lawrence, 2021). Systems based on ASR automatically enforce the requirement that one code switch or assimilate in order to be understood, rejecting the "communicative burden" in which two people will "find a communicative middle ground and foster mutual intelligibility when they are motivated, socially and psychologically, to do so" (Lippi-Green, 2012). By design, then, ASR services operate like people who reject their communicative burden, which Lippi-Green reports is often due to their "negative social evaluation of the accent in question" (Lippi-Green, 2012). As Halcyon Lawrence reports from experience as a speaker of Caribbean English, "to create conditions where accent choice is not negotiable by the speaker is hostile; to impose an accent upon another is violent" (Lawrence, 2021).

Furthermore, we are concerned about discriminatory performance of ASR services because of its potential to create a class of people who are unable to use voice assistants, smart devices, and automatic transcription services. If technologists decide that the only user interface for a smart device will be via voice, a person who is unable to be accurately recognized will be unable to use the device at all. As such, ASR technologies have the potential to *create a new disability*, similar to how print technologies created the print disability "which unites disparate individuals who cannot read printed materials" (Whittaker et al., 2019). The biased performance of ASR, if combined with an assumption that ASR works for everyone, creates a dangerous situation in which those with particular English language accents may find themselves unable to obtain ASR service.

The consequences for someone lacking the ability to obtain reliable ASR may range from inconvenient to dangerous. Serious medical errors may result from incorrect transcription of physician's notes Zhou et al. (2018), which are increasingly transcribed by ASR. There is, currently, an alarmingly high rate of transcription errors that could result in significant patient consequences, according to physicians who use ASR Goss et al. (2019). Other ASR users could potentially see increased danger: for example for smart wearables that users can use to call for help in an emergency Mrozek et al. (2021); or if one must repeat oneself multiple times when using a voice-controlled navigation system while driving a vehicle (and thus are distracted while driving); or if an ASR is one's only means for controlling one's robotic wheelchair Venkatesan et al. (2021).

Given that English language speakers have a multitude of dialects across the world, it is important to consider the ability of English language ASR services to accurately transcribe the speech of their global users. Given past research results (Tatman, 2017; Meyer et al., 2020) and the United States headquarters of Amazon, Google, and Microsoft, we hypothesize that ASR services will transcribe with less error for people who were born in the United States and whose first language is English. We hypothesize that performance of ASR systems is related to the *age of onset*, the age at which a person first started speaking English, which is known to be highly correlated with perceived accent (Flege et al., 1995; Moyer, 2007; Dollmann et al., 2019). But beyond this, based on the nationalist and neoliberal ways in which language is used to reinforce power, we hypothesize that ASR performance can be explained in part by the power relationship between the United States and speakers' birth countries. That is, for the same age of onset of English and other related covariates among speakers not born in the United States, we expect that speakers born in countries which are political allies of the United States will have ASR performance that is significantly better than those born in nations which are not aligned politically with the United States. This paper tests and validates these hypotheses by utilizing a data set with significantly more speakers, across a large number of first languages and birth countries, than those which have previously been used for the evaluation of English ASR services.

## 2 RELATED WORK

### 2.1 AUDITS OF AUTOMATIC SPEECH RECOGNITION

Our work builds on a small but impactful body of literature investigating the disparities in performance of commercially available English ASR services.

Gender has been inconsistently associated with English ASR performance, significantly better for male speakers over female speakers (Tatman, 2017), female speakers over male speakers (Koenecke et al., 2020; Goldwater et al., 2008), or with no significant performance difference (Tatman & Kasten, 2017; Meyer et al., 2020).

There has been evidence that race and geographic background (especially as it relates to accent and dialect) has impact on ASR performance. Speakers from Scotland were found to have worse English ASR performance than speakers from New Zealand and the United States (Tatman, 2017), while speakers who self-identified as speaking with Indian English accents had transcriptions with higher error rates versus speakers who self-identified as speaking with US English accents (Meyer et al., 2020). Finally, ASR services consistently underperform for Black speakers in comparison to white speakers (Tatman & Kasten, 2017; Koenecke et al., 2020).

Significant research has addressed how to make ASR more robust to accent Liu et al. (2022), e.g., by training accent-based modifications to particular layers of a single model; mapping between the phones of two different accents; or using an adversarial network to separate accent-invariant and -variant features. Unlabelled clustering may be used to find accents that are under-represented; oversampling them can then improve performance Dheram et al. (2022). Our work is to audit rather than to repair ASR disparities.

Multiple aspects of spoken English are affected by the particular accent of the speaker, including both: a) how words are pronounced (Lippi-Green, 2012), and b) what words are used and how sentences are structured. By studying unstructured speech, researchers obtain a view of the discrimination experienced by speakers in transcription accuracy as a function of multiple aspects of accent (Koenecke et al., 2020). This paper presents results from a data set that controls for word choice and sentence structure (Weinberger, 2015) so that we can focus on the impact of word pronunciation on ASR performance.

### 2.2 HOW LANGUAGE IS USED TO CONTROL AND DIVIDE

Language, and in many cases specifically English, has a history of being "standardized" by those in power as a medium through which to exert influence Nee et al. (2022). Examples range from English being used to rank and hierarchize those deemed "other" in the United States (Lippi-Green, 2012) to the deliberate introduction of variations of English in India during British colonization in order to maintain societal hierarchies and divisions (Naregal, 2001). We argue the impact of ASR systems is towards more standardization of the English language, which is part of a history of how standardization of language has been a tool to maintain power.

### 2.3 HOW TECHNOLOGY AND AI SHIFT POWER

Auditing ASR services is just one step in building an understanding of how artificial intelligence (AI) has the potential to either consolidate or shift power in society, both on a local and global scale. Whether it be the feedback loops we observe in predictive policing (Ensign et al., 2018; Richardson et al., 2019), the involvement (and experimentation) of Cambridge Analytica in Kenyan elections (Nyabola, 2018), or the significant portion of Amazon Mechanical Turk crowdwork performed by workers in India (Ross et al., 2010; Difallah et al., 2018), all of these phenomena are a part of the coloniality of power:

> [T]he coloniality of power can be observed in digital structures in the form of socio-cultural imaginations, knowledge systems and ways of developing and using technology which are based on systems, institutions, and values which persist from the past and remain unquestioned in the present (Mohamed et al., 2020).

Through our audit of English ASR services (with recordings from speakers born in many different countries) in combination with a historical analysis of the ways in which language and power have been closely intertwined, we both bring attention to and report evidence demonstrating the coloniality of ASR as it exists today.

# 3 MATERIALS AND METHODS

In this section, we describe our data and procedures for our quantitative study of ASR. To stay relevant with the published research on ASR bias, we select ASR services from the five evaluated in Koenecke et al. (2020). The top three performing ASR services in their extensive tests were Google, Amazon, and Microsoft. All three companies are notable not just as cloud service providers, but in the consumer product space in which their ASR services are implemented as part of their own devices. Recordings were transcribed using the three companies' respective speech-to-text APIs in 2021.

## 3.1 WORD INFORMATION LOST (WIL)

To evaluate the correctness of the ASR service transcriptions against the elicitation paragraph that speakers read, we use a metric specifically designed for the assessment of ASR known as *word information lost* (WIL) (Morris, 2002; Morris et al., 2004). WIL is derived from an information theoretic measure of the mutual information between two sources. In short, for our case, it is a *distance* between the elicitation paragraph and the transcription for a speaker. The WIL is given by:

$$\text{WIL} = 1 - \frac{H^2}{(H + S + D)(H + S + I)}, \tag{1}$$

where $H$ is the number of hits, $D$ is the number of deletions, $I$ is the number of insertions, and $S$ is the number of substitutions between the elicitation paragraph and the transcription.

Compared to another commonly used metric, *word error rate* (WER), WIL offers distinct advantages:

1. WIL is defined from 0 (all information preserved) to 1 (no information preserved), whereas WER is similarly lower bounded by 0 but has no upper bound.
2. WIL is symmetric between deletions and insertions, unlike WER, which, especially at high error rates, weights insertions more than deletions (Morris, 2002; Morris et al., 2004).
3. The inaccuracies of WER are more severe at higher vs. lower error rates (Morris, 2002), which can be problematic in linear regression studies.

Particularly in the context of our transcription task and the resulting analyses, the advantages of WIL make it the better metric by which to compare ASR performance.

## 3.2 SPEECH ACCENT ARCHIVE

Our recordings come from *The Speech Accent Archive*, a collection of recordings of speakers born across the world and with different first languages all reading the same text (Weinberger, 2015). Full details on the methodology used in the recording collection and processing are available in Section B in the Appendix. After answering demographic questions, speakers were presented with the elicitation paragraph in Section 3.2.1 and allowed to ask questions about words they did not understand before reading the paragraph once for the recording.

### 3.2.1 ELICITATION PARAGRAPH

The elicitation paragraph below was crafted by linguists to include many of the sounds and most of the consonants, vowels, and clusters that are common to English (Weinberger, 2015).

> Please call Stella. Ask her to bring these things with her from the store: Six spoons of fresh snow peas, five thick slabs of blue cheese, and maybe a snack for her brother Bob. We also need a small plastic snake and a big toy frog for the

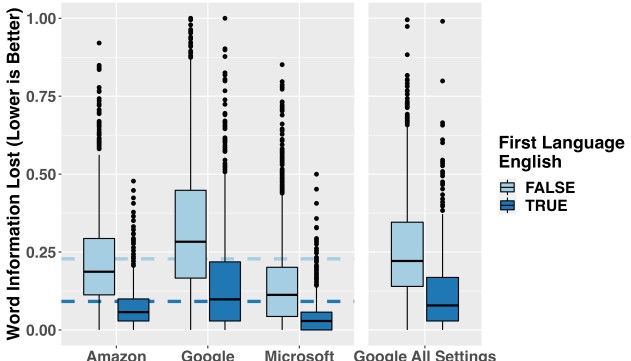

Figure 1: Word information lost by ASR service and English as the speaker's first language. For each service, WIL is significantly lower when English is a first language, with mean performance across all services for each group represented by the dashed lines. Allowing Google ASR to use the "best" dialect setting for each speaker (*Google All Settings*) reduces WIL, but does not remove the significant difference between first and non-first language English speakers.

> kids. She can scoop these things into three red bags, and we will go meet her Wednesday at the train station.

The methodology for recording, demographic information collected, and careful construction of the elicitation paragraph means that *The Speech Accent Archive* contains information particularly well-suited to analyzing how English ASR services perform across a global population.

Further, the use of a constant text allows us to produce results that control for particular aspects of accent. Since all speakers read the same paragraph, any disparity in ASR performance will not be a result of word choice or sentence structure or length — heterogeneities in these may complicate ASR disparity analysis Liu et al. (2022). We can use this to narrow in on ASR disparities that result from the manner of speaking the same words across different English language accents.

### 3.2.2 SPEAKER INFORMATION COLLECTED

The information on speakers collected at the time of recording includes their age, sex (recorded, unfortunately, as a single binary male/female variable), country of birth, first language, age of onset of English speaking, whether they had lived in an Englishs-speaking country, and if so, for how long, and whether the speaker's English learning environment was academic or naturalistic. Age of onset is particularly useful, as it has been shown to be correlated with perceived accent (Flege et al., 1995; Moyer, 2007; Dollmann et al., 2019). This speaker-level information is integrated into the regression performed in Section 4.2.

### 3.2.3 DATA DESCRIPTION

The data set includes 2,713 speakers with an average age of 32.6 years, and an average age of onset of English speaking of 8.9. The speakers represent 212 first languages across 171 birth countries. Figure 2 lists the top ten first languages represented in our data set by the number of speakers.

We note that at the time of recording, 2,023 (74.6%) speakers were either current or previous residents of the United States. By default, most ASR services that would be used on and by these speakers while they are in the United States would likely be configured to use the United States English dialect for transcription. For some of our results, we also use this dialect as the default. In addition, in Section 4.3, we conduct analyses using the "best of" all transcription service dialect settings, and show that the results are primarily the same.

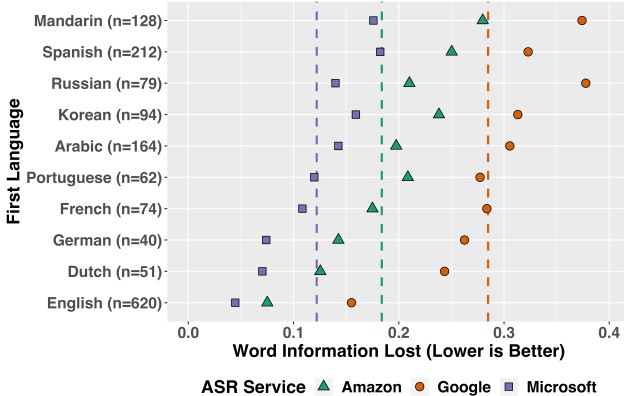

Figure 2: Mean word information lost (WIL) for ASR services vs. first language (showing the top 10 first languages by number of observations, sorted by total mean WIL). Mean WIL performance across all speakers for a specific ASR service is shown as a vertical dashed line and shows service performance from best to worst, is Microsoft, Amazon, and Google.

## 4 RESULTS

### 4.1 GROUP-LEVEL ANALYSIS

In Figure 1, we compare WIL across ASR services grouped by whether a speaker's first language was English. While overall performance differs between services with Microsoft performing best followed by Amazon and then Google, all services performed significantly better ($P < 0.001$) for speakers whose first languages was English. On average across all services, WIL was 0.14 lower for first language English speakers. By service, the size disparities followed overall performance, with a difference of 0.17, 0.14, and 0.10 for Google, Amazon, and Microsoft, respectively.

In Figure 2, we highlight mean ASR performance for the ten first languages for which we have the most data. The order of performance found in Figure 1 is maintained across services — across all ten first languages, Microsoft performs the best, followed by Amazon, and then Google. We find that all the services perform best for those whose first language is English, followed by Dutch and German. The worst performance is on speakers whose first languages are Mandarin and Spanish.

### 4.2 SPEAKER-LEVEL REGRESSION

Motivated by the results in Section 4.1, we construct a linear regression to understand what factors have a significant effect on the performance of ASR services. As discussed in Section 1, the way a person speaks English has and continues to be a basis for discrimination by those in power, and so we include covariates to understand how this discrimination may transfer to ASR services. We want to know if ASR performance is correlated with how the speaker is perceived from a lens of United States global political power. As a broad single measure for this political power, we encode if the speaker's birth country is a part of the North Atlantic Treaty Organization (NATO) as of January 2022.

Specifically, we include the following covariates for each speaker: age; age of onset of English speaking; sex; English learning environment; if their first language is Germanic (as a measure of first language similarity to English, with the list of Germanic languages from Glottolog (Hammarström et al., 2021)); if their birth country is a part of NATO; if they have ever lived in an English-speaking country, and if so (as a nested variable), for how long. We also create nested covariates for English and the United States in the Germanic first language and birth country in NATO covariates respectively to separate the effects of English and the United States specifically.

In order to satisfy the assumptions for linear regression, in particular the normality of the residuals, we perform a square root transform on our response variable, WIL. The diagnostic plots for the regression assumptions can be found in Section C in the Appendix.

### 4.2.1 REGRESSION RESULTS

The results of the regression are shown in Table 1 in Section A of the Appendix under the headings *Amazon*, *Google*, and *Microsoft*. We find multiple covariates that have a significant effect across all three services ($P < 0.05$). Highlights of these findings include:

- WIL increases with a later age of onset of English speaking. As described in Section 3.2, age of onset is correlated with perceived accent.

- WIL decreases with speaking a Germanic first language, having controlled for the effect on WIL of English as a first language, which is also significant.

- Having lived in an English-speaking country has a negative effect on WIL, as does the number of years spent living an English-speaking country.

- Finally, being born in a country that is a part of NATO but is not the United States is associated with a lower WIL.

The final result suggests that a person's birth in a country proximate to the United States' geopolitical power is related to how ASR services perform on their speech.

Some covariates are only significant for certain services - ASR services from Amazon and Microsoft perform significantly worse on males than females. Google and Microsoft perform significantly better for those born in the United States, while Amazon and Google perform significantly better for those who learned English in a naturalistic environment rather than an academic one.

### 4.3 TRANSCRIPTIONS USING OTHER ENGLISH SETTINGS

As explained in Section 3.2.3, a majority (74.6%) of the speakers in the data set were or had been residents of the United States at the time of recording. Thus, we used the United States English setting of all of the ASR services, as this was likely the settings which would be used on or by them.

However, ASR services do offer more settings for English. It is reasonable to ask how much using all of the English language settings available for a transcription service could improve WIL. We decided to understand this question for the service with the worst overall performance and largest disparity in performance, Google, as shown in Figure 1.

We transcribed the recordings using all available English settings that Google supported. Specifically, we try these English settings on Google's ASR service: Australia, Canada, Ghana, Hong Kong, India, Ireland, Kenya, New Zealand, Nigeria, Pakistan, Philippines, Singapore, South Africa, Tanzania, United Kingdom and the United States. To give Google the best opportunity for improvement, for each speaker we took the lowest WIL across all settings' transcriptions. Note that while this is guaranteed to offer the largest improvement, it is unrealistic to do in practice, since it requires knowledge of the ground-truth transcript. We refer to this as *Google All Settings*.

A comparison to Google's original performance is offered in Figure 1, where *Google* refers to transcriptions generated only using the United States setting, and *Google All Settings* refers to WIL generated using the method above. Originally, we saw that for *Google*, first language English speakers had a WIL that was on average 0.17 lower than first language non-English speakers. When using the technique for *Google All Settings* on both first language English and non-English speakers, we notice that a disparity of a similar size (0.14, $P < 0.001$) still exists. In fact, even when we only use the United States English setting for speakers with a first language of English and allow first language non-English speakers to take the lowest WIL from all settings, the disparity is still a considerable 0.10 ($P < 0.001$).

We also note that the significance of the factors in the linear regression did not change when compared to Google's original transcription performance. This result is displayed in Table 1 in Section A of the Appendix under the column *Google All Settings*. This suggests that even Google's attempts to adapt their technology to different global settings are subject to the same biases we highlighted originally.

## 5 DISCUSSION

Across all three ASR services tested (Amazon, Google, and Microsoft), we find significant disparities in performance between those whose first language is English and those whose first language is not English. Moreover, we find that these disparities are connected not only to the age at which an individual began speaking English, the environment they learned it in, and whether or not their first language was Germanic, but also whether or not their birth country is a part of NATO, a representation of political alignment with the United States. When, with one of the services tested (Google), we again transcribed recordings using all of the available English language locality settings, we saw all of our significant results remain the same, implying that the current set of international English language models offered does not solve the inherent problems of bias we observe in ASR.

### 5.1 HISTORICAL CONTEXT

In many ways, we are not surprised by the ways in which a neoliberal capitalist power structure provides better services to a select group of English language speakers. In many ways, it parallels historical colonial use of language, first to standardize language for the benefit of those in power, and second to benefit from hierarchies in language dialects.

Many of today's globally dominant languages were imposed violently within its nation of origin and then the colonial encounter across the world. For instance, the French language was used to turn "peasants into Frenchmen" (Weber, 1976). Within France, French standardization served the dominant class, suppressed the culture of many of its own people, and worked to discipline labor and extract greater profit. ASR systems similarly serve to standardize language — any accent deemed nonstandard is not understood. Speakers are compelled to mimic the dominant accent, which is felt as coerced or even violent (Lawrence, 2021). Providers don't wait to ensure that their ASR product works well across dialects before deploying it – that would break the "move fast and break things" rule and allow competitors to establish market dominance Hicks (2021).

Second, the notion of a "standard" language dialect *enables* social disparity. Education policy during the US occupation of the Philippines enforced racialization; English language lessons were used to emphasize the inferior accent, hygiene and cleanliness of the lower status "Oriental" students (McElhinny & Heller, 2020). In India, British education policies deliberately taught different English dialects to different castes to grant cultural and social authority to a class of elites that aided British governance. Dominant castes entered English medium schools and were tutored by people from England (Gauba, 1974), and were expected to "refine" the "vernacular" languages and teach those to other Indians. They did so while securing their own privileged access to colonial English (Chandra, 2012) and the state power that came with it as the favoured language of governance. The elite monopolized the "correct" language by denying it to other Indians, and thereby made the language selective and exclusive.

ASR services are similarly exclusive; literally providing less control of systems to those with disfavoured accents, and less capability for those in professions which use ASR transcription, e.g., medical professionals. Exclusivity serves an important economic purpose by allowing providers to market beyond the commodity that ASR really is. In this case, it enables marketing of the identity people can have by using it, including the identity of speaking "correct" (white U.S.) English.

### 5.2 FORESIGHT FOR FUTURE ASR DEVELOPMENT

We provide one example of how the lessons from the historical use of language for dominance may apply to discussions of how ASR will evolve in the future. Academic researchers and industry service providers have made claims that the problems are temporary. For example, Amazon claims "As more people speak to Alexa, and with various accents, Alexa's understanding will improve" (Harwell, 2018). As another example, researchers state about the ASR racial performance gap: "The likely cause of this shortcoming is insufficient audio data from black speakers when training the models" (Koenecke et al., 2020).

However, historical hindsight has not indicated that ASR services will improve over time without deeper structural changes. First, by selling products that work for one accent above others, technology companies make speakers of other accents less likely to use their products. Members of groups

historically subject to disproportionate state surveillance may be more hesitant to consent to contribute data towards AI technologies (Jo & Gebru, 2020). Both problems operate as a feedback loop to keep disfavored speakers out of future ASR training data. Further, ML algorithms may naturally tend to discriminate against a smaller group in the population because sacrificing performance on that group may allow reducing average "cost" on the population as a whole, even if the training data represents them in proportion to their population (Zou & Schiebinger, 2018). Finally, ground truth labelling is likely to be less accurate for members a disfavored group, and incorrect labels will be fed back into training of future systems (Denton, 2019). Whittaker et al. describe an example in computer vision for autonomous vehicles — the more video of people in wheelchairs used in training, the *less* likely it was to label a person backing their wheelchair across the street at a crosswalk as a person (Whittaker et al., 2019). Further, in speaker verification, Hutiri and Ding lay out how multiple layers of bias contribute to performance disparities Hutiri & Ding (2022). Instead, historical hindsight indicates that the problem in ASR is more systematic, related to the more fundamental nature of the use of standardized language to divide and provide the benefits of control.

In short, the techno-optimist idea that ASR accent bias will resolve itself in time is unconvincing. Similar to how equity and justice should be centered in each layer of linguistic structure for equitable NLP design Nee et al. (2022), active work will be required to design ASR services that repair damage caused by colonial and post-colonial uses of language and accent to discriminate.

## 5.3 CONCLUSION

This paper extends the results reported in prior English language ASR performance audits. In part, we provide an audit of ASR using a much larger data set containing speech from a large number of countries of birth as well as a large number of first languages. The quantitative results show how ASR services perform on speakers whose first language is English vs. those for which it is not, and how ASR services perform compared to each other. More critically, we find that, controlling for several related covariates about first language, all ASR services perform significantly worse if the speaker was born outside of a NATO country; in effect, in a country further from United States geopolitical power. We argue that this has historical parallel in the ways in which language has been used historically to maintain global power. By explaining these parallels, and by providing quantitative evidence for the effect, we hope that researchers and developers hoping to reduce disparities in ASR services will be better able to identify the systematic nature of the problems.

## 6 ETHICS STATEMENT

While the creation and continued upkeep of *The Speech Accent Archive* does not fall within the scope of this work, we note that all subjects did sign an informed consent form before being recorded, available at `https://accent.gmu.edu/pdfs/consent.pdf`, and that all data used in this work was anonymized.

It is important to recognize that the data set used in this study overrepresents some groups/backgrounds and underrepresents others, while also not including information on other potential influencing factors such as socioeconomic status. One area of focus for future data collection could be speakers who do not currently reside in the United States.

## 7 REPRODUCIBILITY STATEMENT

The recordings and associated demographic data used in this experiment are available upon request from the maintainers of *The Speech Accent Archive* (Weinberger, 2015), or at `https://accent.gmu.edu/`. Section 3 describes the data set and error metric used in our analysis, while Section B in the Appendix describes the data pipeline, including the steps of recording collection, submission to ASR services, transcript processing, and computation of the error rate. The scripts used in the cleaning and analysis of the data will be hosted on GitHub upon publication of the paper.

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

# A    REGRESSION TABLE

Table 1: Speaker-level regression

| | Amazon | Google | Microsoft | Google All Settings |
|---|---|---|---|---|
| | *Dependent Variable:* | | | |
| | Square Root of Word Information Lost (WIL) | | | |
| Age at Time of Recording | 0.001* | 0.002* | 0.001* | 0.001* |
| | (0.0003) | (0.0004) | (0.0004) | (0.0004) |
| Age of Onset of English Speaking | 0.006* | 0.004* | 0.006* | 0.005* |
| | (0.0005) | (0.001) | (0.001) | (0.001) |
| Male | 0.015* | 0.001 | 0.022* | 0.005 |
| | (0.005) | (0.007) | (0.006) | (0.006) |
| Naturalistic Learning Environment | −0.028* | −0.042* | −0.018 | −0.029* |
| | (0.009) | (0.012) | (0.010) | (0.010) |
| Unknown Learning Environment | −0.135 | −0.007 | −0.061 | −0.008 |
| | (0.081) | (0.110) | (0.095) | (0.093) |
| Germanic First Language | −0.074* | −0.082* | −0.074* | −0.071* |
| | (0.012) | (0.017) | (0.014) | (0.014) |
| First Language English [Germanic First Language] | 0.046* | 0.124* | 0.058* | 0.084* |
| | (0.017) | (0.024) | (0.020) | (0.020) |
| Birth Country in NATO | −0.060* | −0.041* | −0.063* | −0.040* |
| | (0.007) | (0.010) | (0.009) | (0.008) |
| Birth Country USA [Birth Country in NATO] | −0.003 | −0.135* | −0.027* | −0.076* |
| | (0.012) | (0.016) | (0.014) | (0.013) |
| Lived in English-Speaking Country | −0.035* | −0.053* | −0.044* | −0.057* |
| | (0.009) | (0.012) | (0.010) | (0.010) |
| Years in English-Speaking Country [Lived in English-Speaking Country] | −0.002* | −0.002* | −0.001* | −0.002* |
| | (0.0003) | (0.0005) | (0.0004) | (0.0004) |
| Intercept | 0.400* | 0.496* | 0.283* | 0.454* |
| | (0.012) | (0.016) | (0.014) | (0.013) |
| Observations | 2,713 | 2,713 | 2,713 | 2,713 |
| $R^2$ | 0.332 | 0.258 | 0.266 | 0.274 |
| Adjusted $R^2$ | 0.329 | 0.255 | 0.263 | 0.271 |
| Residual Std. Error (df = 2,701) | 0.139 | 0.190 | 0.164 | 0.160 |
| F Statistic (df = 11; 2,701) | 121.930* | 85.259* | 88.773* | 92.863* |

*Reference Classes: Female & Academic Learning Environment*          $^*P < 0.05$

# B    SPEECH ACCENT ARCHIVE RECORDING AND PROCESSING

## B.1    DATA COLLECTION

The following information about the data collection process comes from *The Speech Accent Archive* (Weinberger, 2015).

Subjects were sat 8-10 inches from the microphone and recorded individually in a quiet room. They were each asked the following questions:

- Where were you born?
- What is your native language?
- What other languages besides English and your native language do you know?
- How old are you?
- How old were you when you first began to study English?
- How did you learn English (academically or naturalistically)?
- How long have you lived in an English-speaking country? Which country?

Subjects were asked to look over the elicitation paragraph and ask questions about any unfamiliar words. Finally, they read the passage once into a high-quality recording device.

## B.2 Data Processing

All recordings were initially converted into the mp3 file format and then subsequently converted into the formats necessary for transcription by each of the respective services. This was done to help control any effects which might arise from files being originally recorded in lossy instead of lossless formats.

Audio files were submitted to the respective APIs for all three service providers and the returned transcripts were then concatenated into a single string for each speaker. Across all services, only once did a service fail to return a transcription, and this occurred only for a specific triplet of service, speaker, and transcription dialect. Transcripts were then cleaned using the following process:

1. Semicolons were converted to spaces.
2. Characters were converted to lowercase.
3. Hyphens and forward and back slashes were replaced with spaces.
4. All currency symbols, ampersands, equals signs, octothorpes, and percent signs were separated by spaces on both sides.
5. The string was split on spaces to create words.
6. Punctuation at the beginning and end of words was replaced with spaces.
7. Leading and trailing spaces were stripped.
8. Words that were only spaces were deleted.
9. Words exactly equal to the characters "3", "5", and "6" were converted to "three", "five", and "six", since these exact numbers appear in the elicitation paragraph as written in Section 3.2.1 and would be correct transcriptions.
10. Spaces were added back in between words and recombined into one string.

After putting all transcripts and the elicitation paragraph through this process, WIL was calculated using the jiwer Python package (Morris et al., 2004).

## C  Checking Regression Assumptions

Before looking at the results of our regression, we evaluate the regression assumptions via diagnostic plots in Figures 3, 4, 5, and 6. Due to the square root transform which we performed on WIL (our response variable) in Section 4.2, the diagnostic plots show that our regression assumptions are satisfied, although there are some outliers to investigate. We analyzed each labelled outlier from the plots by hand, first by checking the speaker data to make sure there are no anomalies, and then by listening to the recording to ensure there are no audio issues. Having done this, we proceed to interpret the results of the regression as described in Table 1.

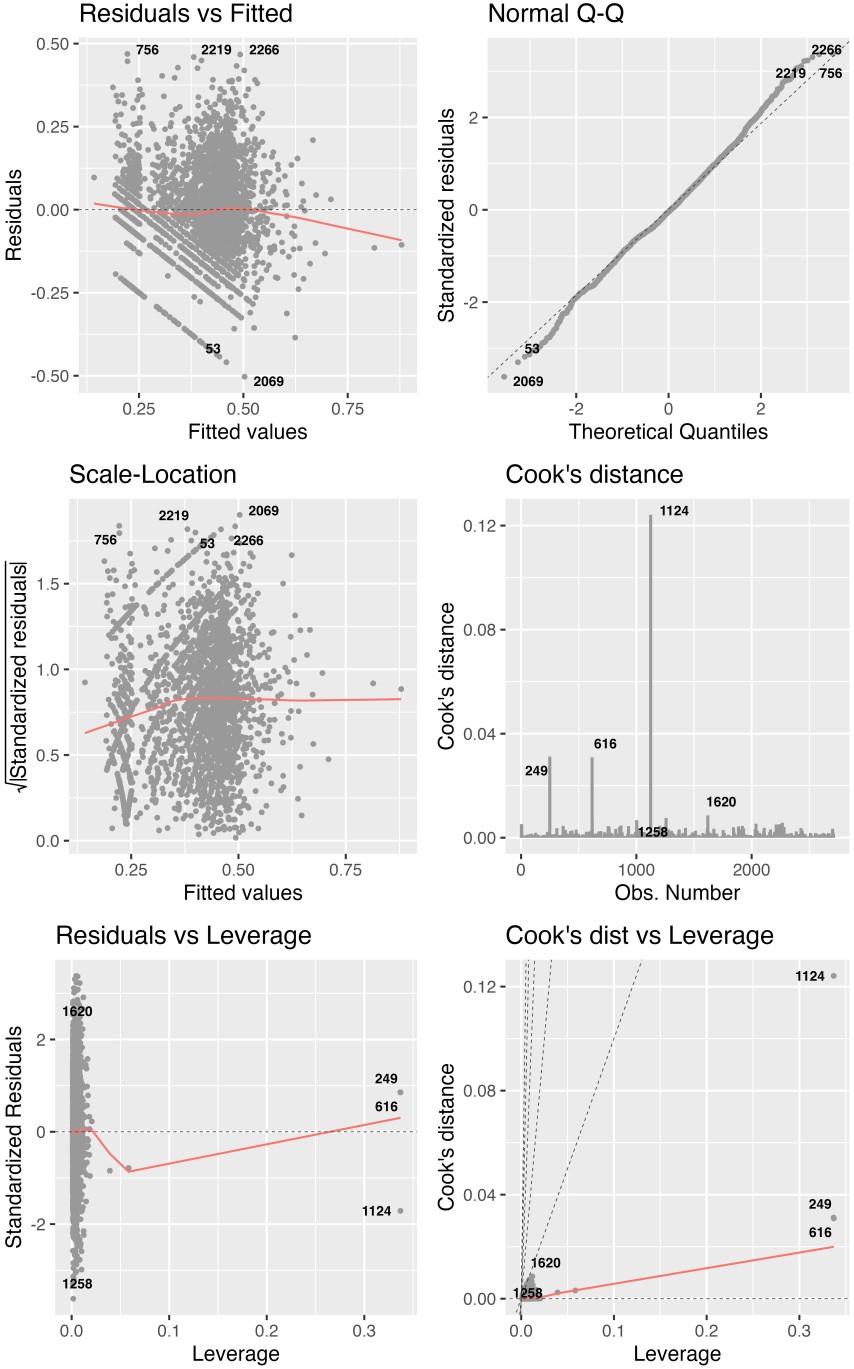

Figure 3: Checking the regression assumptions for Amazon

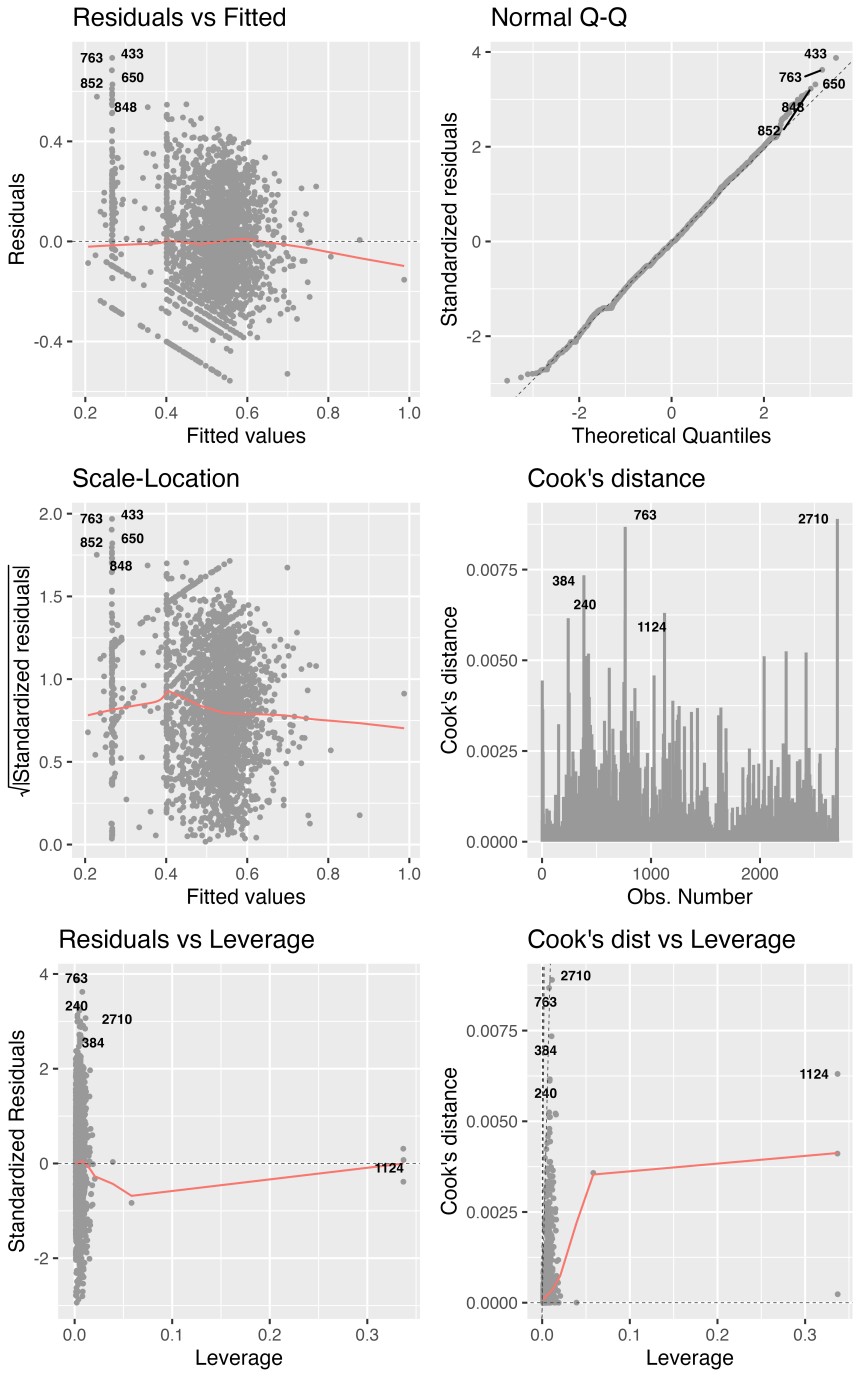

Figure 4: Checking the regression assumptions for Google

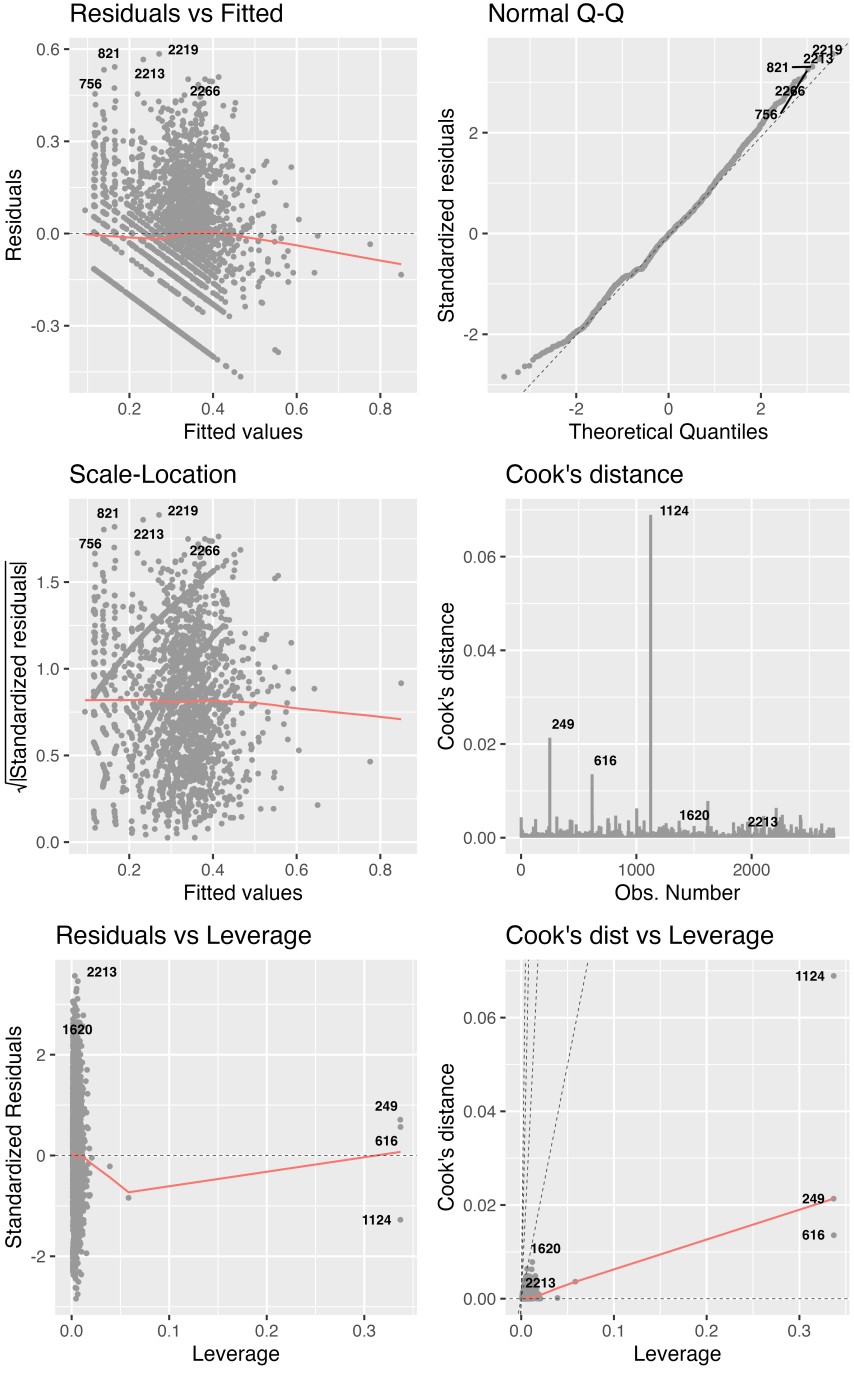

Figure 5: Checking the regression assumptions for Microsoft

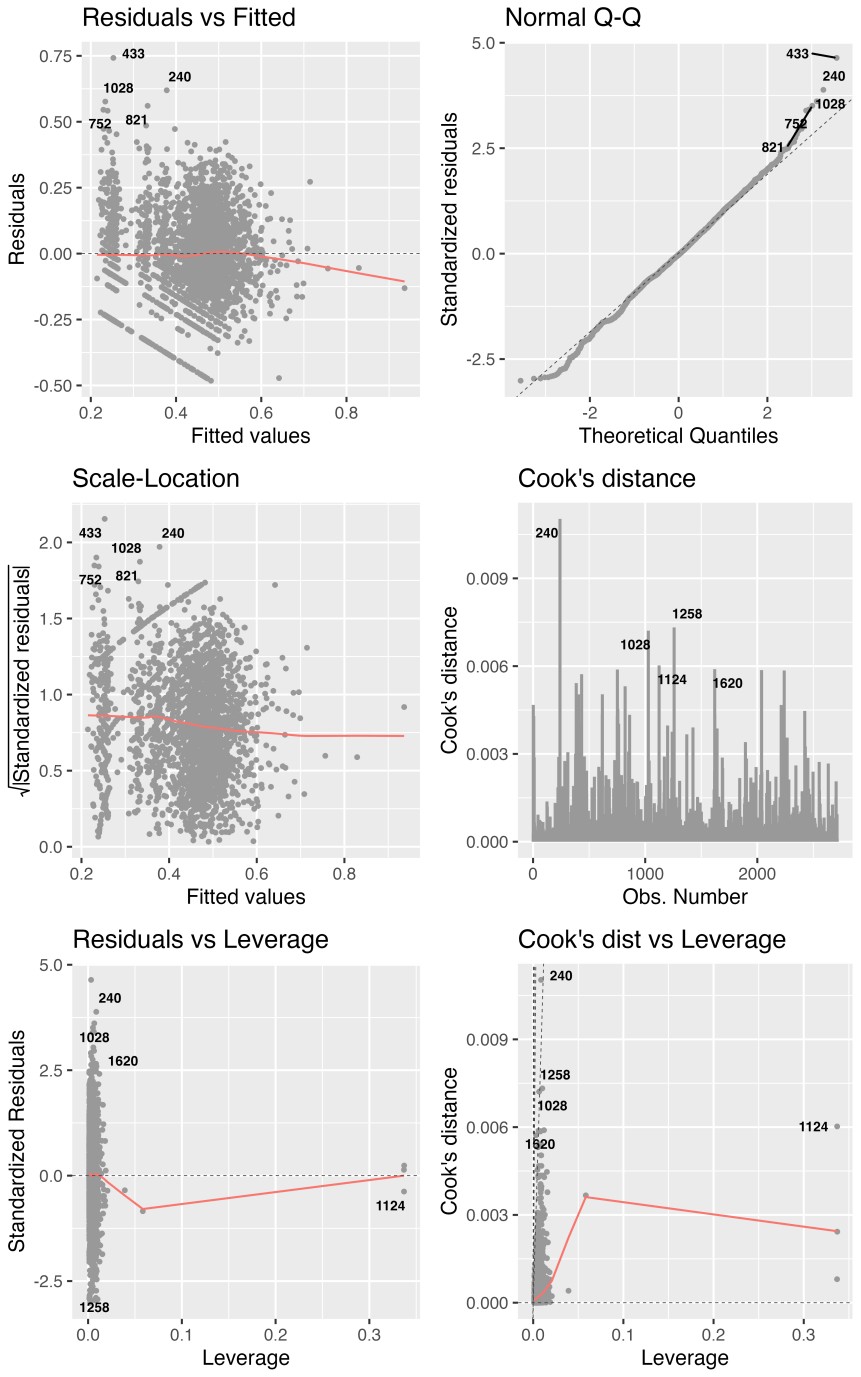

Figure 6: Checking the regression assumptions for Google All Settings

