# OpenReview forum: "Performance Disparities Between Accents in Automatic Speech Recognition"
_ICLR.cc/2023/Conference — Submitted to ICLR 2023_

### Official Review · Reviewer_fz1F · 2022-10-24

**Confidence:** 4
**Correctness:** 3
**Technical Novelty And Significance:** 2
**Empirical Novelty And Significance:** Not applicable
**Recommendation:** 3

**Clarity, Quality, Novelty And Reproducibility:**

The motivations and investigation details are clear.
This paper is a high-quality investigation report of the bias of existing asr systems.
Could not find much novelty parts in terms of technical part.

Detailed comments and questions:

1 Existing systems are bias – not by design – but by the amount of available data for training asr models.


**Strength And Weaknesses:**

Strong:

1 Biases were found in existing commercial systems such as amazon, google and microsoft’s asr systems for voices from different regions and different accents;

Weak:

1 Could not find technical novel parts and this paper is an investigation report without detailed technical solutions.


**Summary Of The Paper:**

This paper evaluates the performance differences among accents in asr systems such as amazon, google and microsoft’s asr systems. The systems have biased word error rates when evaluated by a large and global dataset of speech from the speech accent archive.
This is more like an investigation report, instead of a reach paper with novel methodology or technical solutions.


**Summary Of The Review:**

Could not find novel ideas or technical solutions. This is an investigation report.
Can not give a high recommendation score currently.

---

> ### Author Response · Authors · 2022-11-18
> **Reply to Reviewer fz1F**
>
> The reviewers may certainly accept or reject our submission, we accept their decision.
>
> *However, we request that you reconsider your use of “technical” or “novel” in this case*.  Our work is novel; no published work has identified and provided quantitative evidence regarding the geopolitical power aspect of the bias in current ASR systems. This is an issue of growing importance globally for speakers of English, of whom there are more than a billion people in the world.
>
> The ICLR 2023 CFP lists “societal considerations of representation learning including fairness” as one of 15 relevant topics, *which is why we chose to submit here*, and why we believe that reporting on societal considerations of ASR algorithms, including their fairness of performance among global speakers of English, is a technically relevant topic for ICLR 2023.
>
> This work is “technical” as many papers in CS are; we use what technical tools are necessary to understand and model the necessary system issues. These technical tools include those from linguistics, history, power and oppression, without which we could not have generated the hypothesis which we test in this paper. Then we apply technical tools from linguistics and statistics to perform a black-box bias evaluation of multiple top ASR services. These are all technical tools. We use the right tools to advance the study of ASR.  Consider other applications of technical tools: when researchers need to apply physics to improve motion models; or apply biology to improve medical imaging; we accept that as “technical”.  So why do we decide that application of history, linguistics, and power and oppression is not “technical” when it is needed to produce a “high-quality investigation report of the bias of existing asr systems”?  The plain truth is that the application of tools from these fields *is* technical, is challenging, and requires a novel combination of interdisciplinary expertise, which we have assembled and used in this submission to make a concrete contribution that matters to people in the world who speak English.

---

### Official Review · Reviewer_D6XQ · 2022-10-25

**Confidence:** 3
**Correctness:** 2
**Technical Novelty And Significance:** 3
**Empirical Novelty And Significance:** 2
**Recommendation:** 5

**Clarity, Quality, Novelty And Reproducibility:**

The paper is clearly written. Novelty of WIL and its correlation with various aspects of L2 English acquisition e.g. age, Germanic root, etc. are not particularly novel or interesting.

The hypothesis that the disparity of various commercial recognizers is related to political power isn't well researched, ie given other possible explanations that are neither suggested or explored as noted above. There are also many statements that aren't fully backed and tend to be of an opinionated nature, e.g.
- "assumption that ASR works for everyone", citation?
- "dangerous situation" "particular English language accents ... unable to obtain basic services." - this seems rather exaggerated. in what situations would this disparity result in lack of basic service or a dangerous situation?
- "attempted standardization of the English language via the increasing ubiquity of ASR systems is another chapter in a long story of how movements create a standard language... have been tools to maintain power." - a clear link between a disparity in accented vs native ASR performance as a tool to maintain power hasn't been established, how can it be described in this manner?

The work is likely reproducible. The source data from Speech Accent Archive is available on request, but at the moment none of the resulting ASR transcripts or processing scripts are available. Having an anonymized github source would be useful in reviewing the results especially given the conclusions being drawn.

**Strength And Weaknesses:**

Strengths
- Performs a controlled study of how various commercial recognizers perform with a fixed text that is read by speakers that have provided demographic information.

Weaknesses:
- Without samples of ASR results from the collection, cannot tell if there were systemic experimental design biases in the results due to poor text/scoring normalization choices. e.g. are numbers normalized, would a 6 vs "six" be penalized?
- WER doesn't necessarily correlate with overall performance; errors on filler or hesitation words are inconsequential, errors on names very problematic. WIL likely has the same problems. Would have been useful to publish WER alongside WIL for those less familiar with WIL.
- Does WIL actually reflect user satisfaction? for example, how do aspects of L2 English acquisition compare on completing Alexa requests or transcribing videos?
- The lack of differences between Google and All-Google settings seems strange: it would be interesting to see a table matrix showing how each of the sub accent systems compare to the EnUs for matched cases, ie. EnIn tested on EnUs vs EnIn on EnIn, EnUk vs EnUs, etc.  why would Google offer multiple accent versions if there is no substantial difference?
- The work uses read speech to analyze performance, however most systems expect spontaneous speech.
https://www.researchgate.net/publication/221999276_Differences_between_acoustic_characteristics_of_spontaneous_and_read_speech_and_their_effects_on_speech_recognition_performance
- the sum of these experimental issues may invalidate the findings, magnify intrinsic biases in the ASR systems, or be inconsequential
- the work fails to explore other possible sources of the disparity / it is too quick to assign political power:
   * is it simply a data issue? do worse performing population simply have less data available to train on? author's could try to train systems with different proportions of accented data
    * is it technical? is modeling of accents more difficult as they deviate further linguistically from English? does the level of linguistic difference matter, e.g. grammatic verb-order vs pron differences?
    * is it historical? do the least well performing demographics come from populations
    * it would be useful to explore and eliminate other possible sources before solely describing power relationship between birth country and united states as the reason for disparities
   * is this something that holds for other power relationships? would native Russian speakers of Chinese do better on Baidu's ASR systems? how about on American Chinese ASR systems?





**Summary Of The Paper:**

The authors present a paper on biases in mainstream commercial speech recognition services from Amazon, Google and Microsoft that:
1. shows how there are statistically significant performance differences in a word information lost (WIL) metric when the speech that is being transcribed comes from a native vs non-native speaker/age of onset of English speaking.
2. speakers that learn English in a naturalistic environment have lower WIL than in an academic
3. WIL is lower for speakers whose first language is a Germanic one
4. there is a statistical significant correlation between WIL and the speaker's birth country's political alignment to the United States.

The findings that speech recognition systems do worse for non-native, academic environment and non-Germanic L1 vs Germanic L1 are all not particularly surprising and I don't believe are novel findings. Reporting WIL to demonstrate this is likely novel. The main contribution of this paper is showing a political aspect to the disparity.

**Summary Of The Review:**

The paper provides unsurprising results showing disparities in recognition performance given various aspects of native vs non-native speakers of English. A political hypothesis to these disparities is novel, however isn't sufficiently explored to be considered well-supported.

---

> ### Author Response · Authors · 2022-11-18
> **Reply to Reviewer D6XQ**
>
> With regards to “biases … normalization choices”: We did normalize numbers as the reviewer asks about.  We have more detailed information on the processing of speech transcripts in the appendix.
>
> With regards to “differences between Google and All-Google settings”: The performance typically does improve with the All-Google setting, as shown in Figure 1. Many of the covariates have a lower amplitude in the regression (for example Birth Country USA and First Language English), as seen in Table 1. But despite the generally better performance, there is no difference in the main result of the paper, that there is a statistically significant correlation in performance with US political power, when controlling for the other factors.  We have emphasized this in the revision.
>
> Thank you for the reference on spontaneous speech. There are other papers that study bias in spontaneous speech, e.g. the bias against African American accents of English studied in [Konecke 2018].  Our study is meant to be complementary, to answer a different question, which we believe helps to further diagnose the problem.
>
> We can’t answer all important questions on “other possible sources” in one paper, we do not have access to the ASR source or training data.  We have added one variable in the Speech Accent Archive in response to reviewer mfgs, but we are limited to those covariates in the Archive.  Our argument for answering the question we did answer, is that it is important that we first test our hypothesis about ASR bias and report the result, in order to help the community to then narrow in on the “possible sources of the disparity”.  Our main finding holds significance across all three major commercial ASR services tested, which gives some confidence about the result. This submission is a first but necessary step.  We feel there is value in our independent evaluation.
>
> Regarding the criticism that “The hypothesis that the disparity … neither suggested or explored as noted above”:  One advances research by testing hypotheses, and in this submission, we test and validate one hypothesis. That allows us to say the disparities in each ASR system is correlated with our measure of US geopolitical power, with statistical significance, when controlling for multiple other specific covariates (the ones linguists recorded in the Speech Accent Archive because they are believed to influence accent).  Thus we have explored possible explanations for ASR performance differences, ones that are used by linguists.  More research is certainly recommended, but again, this submission represents a first but necessary step.
>
> The “assumption that ASR works for everyone” is in a paragraph that is evaluating a “potential” future if ASR is built into systems along with the assumption that ASR works for everyone.  We have clarified this in the revision. Also, this assumption is sometimes used to make policy. E.g., some faculty are required, for video-recorded lectures, to include “99% accurate transcripts or captioning” [Nordmann 2022].  This results in many hours more workload for faculty whose voice is not accurately transcribed by the ASR service.
>
> Being unable to use an accurate voice assistant may be dangerous. Serious medical errors may result from incorrect transcription of physician’s notes [Zhou 2018], which are increasingly transcribed by ASR services.  There is, currently, an alarmingly high rate of transcription errors that could result in significant patient consequences, according to physicians who use ASR [Goss 2019].  Other ASR users could potentially see increased danger: for example for smart wearables that users can use to call for help in an emergency [Mrozek 2021]; or if one must repeat oneself multiple times when using a voice-controlled navigation system while driving a vehicle (and thus are distracted while driving); or if an ASR is one’s only means for controlling one’s robotic wheelchair [Venkatesan 2021]. We have added this into the revised introduction.
>
> Regarding the question, “a clear link between a disparity in accented vs native ASR performance as a tool to maintain power hasn't been established, how can it be described in this manner?”:  We provide references which discuss language standardization as a historical tool to maintain power; the sentence simply says that ASR has an impact in the same direction, and is in the context of this history.  We have rewritten this sentence to clarify this point.  We do want to note that users with disfavored accents, including those who speak “native” English but not with a “standard” American accent, describe their voice assistant’s requiring them to fake a “standard” American accent to receive service as “othering that is biased and disciplinary” [Lawrence 2021].  What are we choosing if we describe these users’ experiences without describing the historical context which shapes their experiences?

---

### Official Review · Reviewer_mfgs · 2022-10-26

**Confidence:** 4
**Correctness:** 3
**Technical Novelty And Significance:** 2
**Empirical Novelty And Significance:** 3
**Recommendation:** 5

**Clarity, Quality, Novelty And Reproducibility:**

The paper is well-written. The empirical analysis has been conducted on a publicly available dataset and the authors promise to release their scripts to ensure reproducibility.

**Strength And Weaknesses:**

The main strength of this work is that it takes a thorough critical look at ASR accent bias. The authors evaluate three popular ASR services by Microsoft, Google and Amazon on accented English speech from The Speech Accent Archive representing 212 first languages across 171 birth countries. They find that ASR performance is significantly better for speakers whose first language was English. While such observations have been previously documented, this work also examines a number of speaker covariates like age of onset of English speaking, environment where English was learned (naturalistic vs. academic), etc. These covariates were found to have a significant effect on ASR performance across all three services.

I consider the following to be the main weaknesses of this work:
1. The covariate referring to whether the speaker's birth country is a part of NATO was found to be associated with lower WILs. Based on this observation, the authors hypothesized that how far a speaker's birth country is from United States' geopolitical power is correlated with how ASR services perform on their speech. This hypothesis appears to be a bit tenuous. It's unclear if there are other hidden variables that might also be playing a role here. For example, the number of years the speaker lived in the US, how often the speaker interacts with native speakers of English, etc.
2. As the authors note, a majority of the speakers in the data set were residents of the United States at the time of recording. It would have made for a compelling story if the authors had created and released a more representative dataset of accented speech without the bias of US residency.
3. The related work section completely ignores the fairly large body of work that focuses on improving ASR for accented speech. For example, please refer to "Accented Speech Recognition: A Survey" by Hinsvark et al., 2021 for a recent survey. Citations to some other recent works on fairness in ASR are also missing. For example, "Model-based approach for measuring the fairness in ASR", Liu et al., 2021 and "Toward Fairness in Speech Recognition: Discovery and mitigation of performance disparities", Dheram et al., 2022.

**Summary Of The Paper:**

In this work, the authors dissect the problem of discriminatory Automatic Speech Recognition (ASR) performance on accented speech along the following three dimensions:
1. By evaluating predominant ASR services on a large and global dataset of English speech in varying accents.
2. By identifying speaker covariates in a linear regression model that significantly impact ASR performance.
3. By examining the quantitative results within the larger context of the role of language in exacerbating inequality and making ASR services more exclusive.

**Summary Of The Review:**

While the main thesis of this work is well-motivated and the paper is written well, in its current form, I think the paper is marginally below the acceptance threshold mainly due to the concerns I have stated above.

---

> ### Author Response · Authors · 2022-11-18
> **Reply to Reviewer mfgs**
>
> Regarding the potential for missing some hidden variable which could play a role in ASR performance, we have updated the regression analysis as follows, based on your suggestion. We cannot tell how many years the speaker has lived in the US, specifically. But the Speech Accent Archive includes whether or not a speaker has lived in an English-speaking country; and if so, for how long.  We have modified our regression analysis for all services to include this variable, and the revision of the paper reflects this update. The inclusion of this additional variable did not substantially change the main result, that a speaker's birth country being part of NATO is associated with lower WIL.
>
> We acknowledge that beyond this, there are more correlations that could be tested, but this is the extent of the information available from the Speech Accent Archive.
>
> It would be valuable, as suggested, to conduct a campaign to create an additional, more globally representative dataset for the purposes of evaluating the relationship between ASR performance and accent by country of birth.  This paper is not the final conclusion, but we feel it also should not be rejected simply because it is based on one dataset.  In particular, the Speech Accent Archive is large, carefully collected, and well regarded by linguists who study English language accent.  This archive includes multiple covariates that linguists hypothesize to be primary variables that influence accent.
>
> Thank you for the references.  We have included the results from those papers in the revised version of our paper.

---

### Decision · Program_Chairs · 2023-01-20

**Decision:**

Reject

**Justification For Why Not Higher Score:**

1.  Performance disparities in terms of accent in ASR is a well-known issue.  A more in-depth discussion about existing works should be included.

2. The supporting evidence to the conclusion needs to be stronger.  The Speech Accent Archive dataset is good but it may not include other factors as suggested by the reviewers to further support the conclusion in this paper.

3.  Some relatively minor issues in the experimental design that need to be taken care of.

**Justification For Why Not Lower Score:**

N/A

**Metareview: Summary, Strengths And Weaknesses:**

In this paper the authors investigate the ASR performance from a number of major ASR services with respect to factors such as countries of birth and first languages.  The investigation is carried out on the Speech Accent Archive dataset which is a large dataset with meta information. With a careful and controlled analysis the authors draw the conclusion that the ASR performance from some major service providers demonstrates a bias towards the political alignment of the speaker's birth country with respect to the United States'  geopolitical power.  The paper is well motivated and well written.  The topic under investigation is important and has its value to the community in terms of  AI fairness.  That being said,  there are numerous concerns raised in the reviews.  First,  performance disparities in terms of accent in ASR is a well-known issue.  A more in-depth discussion about existing works should be included.  Second, the supporting evidence to the conclusion needs to be stronger.  The Speech Accent Archive dataset is good but it may not include other factors as suggested by the reviewers to convincingly support the conclusion in this paper.  There are also numerous other issues in the experimental design that need to be taken care of.   Overall, the paper needs improvements in order to get accepted.  I would suggest the authors take care of the comments by the reviewers and submit to another venue.